

# Modelling the effects of emotional engagement and peer interaction on the continuous intention to use asynchronous e-learning

Sultan Hammad Alshammari and Mohammed Habib Alshammari

Department of Educational Technology, College of Education, University of Ha'il, Hail, Saudi Arabia

## ABSTRACT

Even though asynchronous e-learning has become popular among universities, few studies have examined how students intend to continue using it for their learning. This study proposed a theoretical model that aims to examine the effects of external factors—emotional engagement and peer interaction—and two constructs of technology acceptance model (TAM) on students' continuous intention to use asynchronous e-learning. A total of 259 students participated in a survey. The data were analysed using two steps in SEM AMOS. confirmatory factor analysis was conducted for assessing model measurement, and structural equation modelling was applied for assessing relationships among constructs and testing hypothesis. The results showed that emotional engagement had a significant effect on students' continuous intention to use asynchronous e-learning through the perceived ease of use of asynchronous e-learning and not through its perceived usefulness. Furthermore, peer interaction had a significant effect on students' continuous intention to use asynchronous e-learning. Moreover, the constructs of the technology acceptance model—perceived ease of use and perceived usefulness—had direct and significant effects on students' continuous intention to use asynchronous e-learning. Several implications and suggestions were discussed.

## INTRODUCTION

Recently, universities have been paying more attention to asynchronous e-learning due to its ability to provide learning without restrictions in terms of locations, times and geography (*Kim et al., 2021*). Asynchronous e-learning is an e-learning type which focuses on learner-directed approaches and is suitable for adult learners (*Carstairs et al., 2014*; *Xing et al., 2018*). During asynchronous e-learning, students can learn from content provided in several different formats, such as text, video and audio, which is designed and prepared by their instructors in advance and delivered to them using a type of learning management system (LMS) such as a blackboard. Thus, interactions between instructors and learners can be managed asynchronously. Differing asynchronous e-learning designs and approaches have been reported in the literature, ranging from types that include interactive elements

Corresponding author
Sultan Hammad Alshammari,
sultan9573@hotmail.com

to slide-based models (*Rouleau et al., 2019*). The delivery design also varies and can be categorised into the following types: (1) using e-learning to supplement and enhance learning in face-to-face classes; (2) learning through e-learning and face-to-face classes are combined; (3) pure e-learning is used, offering students full independence without face-to-face classes (*Regmi & Jones, 2020*).

In asynchronous e-learning, students could study courses at any available time that suits them and can select the strategies of learning that they prefer. Therefore, asynchronous e-learning enables students to interact and have extra time for engaging deeply with and reflecting on the materials of learning, that is essential for improving their comprehension, higher order thinking skills and ability to learn new concepts (*Kim et al., 2021*). Even though asynchronous e-learning has huge advantages, it also has some limitations. For instance, some students may feel isolated because of the delay in the interactions between instructors and students and between classmates (*Moody, 2004*). Thus, some students may not be fully engaged with the contents of their courses (*Kim et al., 2021*). These issues could have negative and direct impacts on their learning process.

Students' continuous intention toward using asynchronous e-learning is essential for them to achieve better progression and outcomes from their courses. However, it is still unclear how this intention develops and is sustained among students in universities. Without this knowledge, instructors have difficulties in developing effective online courses for asynchronous e-learning and particularly in implementing suitable strategies for keeping students interested in their asynchronous e-learning courses. Therefore, it is necessary to assess which factors could impact students' continuous intention to use asynchronous e-learning courses.

Emotional engagement, which consists of cognitive and behavioural dimensions (*Fredricks, Blumenfeld & Paris, 2004*), may affect students' intended use of asynchronous e-learning. This type of engagement is related to belonging sense to a learning community, students' relationships with their peers and instructors and the impact of affective dimensions on learning in general (*Fredricks, Blumenfeld & Paris, 2004*). Furthermore, peer interaction could be an influencing factor which may affect students' continuous intention to use asynchronous e-learning. It refers to the support that learners provide to each other in the environment of e-learning (*Goh et al., 2017*). *Lu, Pang & Shadiev (2023)* conducted a study in which they extended TAM to examine the impacts of particular factors—extrinsic and intrinsic motivations, cognitive engagement and multiple types of resources—on students' continuous intention to use asynchronous e-learning. The authors found that cognitive engagement directly affected students' intention to utilise asynchronous e-learning. *Lu, Pang & Shadiev (2023)* recommended extending TAM and examining the effects of emotional engagement and peer interaction on students' continuous intention to use asynchronous e-learning. It is not yet clear whether these two factors have an effect on students' continuous intention to use asynchronous e-learning. Furthermore, to the best of our knowledge, the effects of these two factors on students' continuous intention to use asynchronous e-learning have not yet been assessed using extended TAM.

In order to address these gaps in literature, this study extends TAM and assesses the effects of emotional engagement, peer interaction and TAM constructs on students'

continuous intention to use asynchronous e-learning. This study contributes to literature by proposing an extended TAM. TAM was established by *Davis (1989)* and has been utilised in prior studies to assess the adoption of different technologies (*Oyman, Bal & Ozer, 2022*). However, the extended version of the TAM and the existing constructs of the model used in previous studies were insufficient in identifying the factors influencing students' continuous intention in the asynchronous e-learning context. Accordingly, it is essential to incorporate into the TAM the main factors which may affect students' continuous intention to use asynchronous e-learning and to examine the relationships between these factors.

## Theoretical framework

This study uses TAM as a theorical framework. Technology acceptance means the extent which users intend to utilise and keep using any new technology. TAM is a validated model which has been used to analyse instructors and students' behavioural intention to use new technologies in several different educational contexts, such as teachers' intention with using games (*Yeo, Rutherford & Campbell, 2022*), LMS (*Alshammari, 2020*) and mobile-based assistive technology in learning languages (*Hsu & Lin, 2022*).

The TAM theoretical framework contains four main factors: perceived usefulness (PU), ease of use (PEU), attitude (ATT) and the behaviour intention to use a technology (BI). PEU means the extent which users feel that using particular technology will be easy and require little effort (*Davis, 1989*). PU means the extent which users believe that using specific technology could be beneficial and could enhance their tasks and performance (*Davis, 1989*). Furthermore, ATT refers to a user's negative or positive feelings regarding using a technology. It was recommended that the attitude construct should be eliminated from the TAM due to the direct effects of PU and PEU on users' intention (*Davis, 1989*). BI is defined as the intention of users toward using particular technology or application (*Davis, 1989*).

## Emotional engagement with TAM

Emotional engagement refers to students' emotional interaction with their learning in either a positive sense (such as when learning aligns with students' identities, satisfaction, interests and values) or a negative sense (for instance, when learning leads to anxiety) (*Chan, Kam & Wong, 2022*). Several previous studies revealed that, emotional engagement showed to be as indicator for students' high academic achievement. *Kuo et al. (2014)* determined that increasing students' emotional engagement led to improved learning outcomes. Similarly, *Kahu & Nelson (2018)* established that emotional engagement could predict the academic success of students. Furthermore, *Wang et al. (2015)* assessed the relationships between emotional engagement in LMSs and students' academic performance and found that emotional engagement with learning activities through LMSs had a direct influence on students' academic and learning performance. However, emotional engagement's effect on students' continuous intention for using asynchronous e-learning remains unknown. Based on this analysis of the literature, hypothesis is formulated as following:

H1: Emotional engagement affects students' perceptions of the usefulness of asynchronous e-learning.

H2: Emotional engagement affects students' perceptions of the ease of use of asynchronous e-learning.

## Peer interaction with TAM

Peer interaction means the interactions between learners with their instructors or with other learners. When students interact with other students and instructors, they can improve their ability to build knowledge (*Liaw, Huang & Chen, 2007*). *Liu et al. (2010)* examined the constructs which may influence students' utilises of e-learning communities. They found that the peer interaction of learners and the perceived ease of online learning community had affected students' use of that LMS. Similarly, Rose et al. (2015) determined that the peer interaction of learners affected students' intention to utilise LMS. Thus, the hypothesis is formulated as following:

H3: Peer interaction affects students' perceptions of the usefulness of asynchronous e-learning.

H4: Peer interaction affects students' perceptions of the ease of use of asynchronous e-learning.

## TAM constructs

Regarding TAM constructs, previous studies have shown that students' perceived ease of using particular technology directly affect their perceptions of the usefulness of that technology. Furthermore, PU and PEU affect the intention of using a technology. These results have been confirmed in several contexts, such as in language learning using mobile phones (*Hsu & Lin, 2022*), LMS (Alshammari, 2018) and LMSs (*Yalcin & Kutlu, 2019*). Accordingly, hypothesis is formulated as following:

H5: Perceived ease of use affects students' perceptions of the usefulness of asynchronous e-learning.

H6: Perceived ease of use affects students' continuous intention to use asynchronous e-learning.

H7: Perceived usefulness affects students' continuous intention to use asynchronous e-learning.

Figure 1 shows the proposed mode.

## Methodology
### Research design

A quantitative design approach is used in this study and data was collected by questionnaires. Quantitative research aims to explain different phenomena by examining numerical data that is analysed using a mathematical technique, specifically, a statistical approach (*Creswell, 2013*). The quantitative research design is systematic, formal and objective and aims to assess effects, causes and relationships between variables (*Queirós, Faria & Almeida, 2017*). Thus, design of quantitative research is suitable for this research, since the study aim at proposing a theoretically extended TAM and exploring the effects of external factors (namely, emotional engagement and peer interaction) and TAM constructs on students' continuous intention to use asynchronous e-learning.

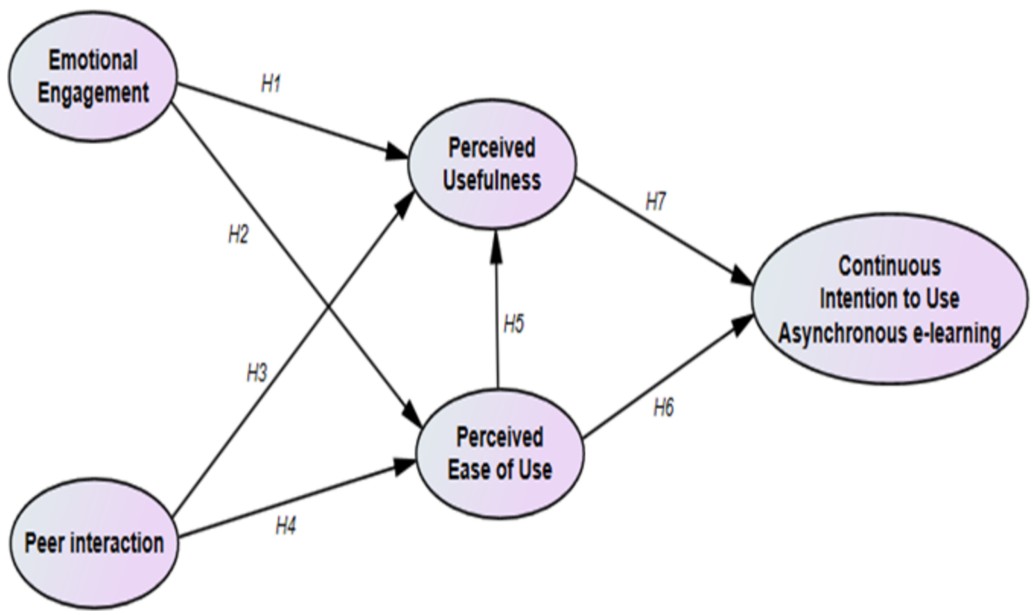

**Figure 1  Proposed model.**

## Instrument

The questionnaire used in this study had two main parts. The first part was designed by the author and aimed to collect data regarding students' demographic information, such as their gender, academic programme, major and college. The second part had 18 items which measure all constructs of the proposed model. All the items that assessed variables were adapted from prior studies. For instance, nine items which measured TAM constructs were utilised from a study of *Davis (1989)*. From *Wara, Aloka & Odongo (2018)*, six items that measured emotional engagement were adapted, and the three items that measured peer interaction were adapted from a study by *Yang & Chang (2012)*.

## Data collection and participants

In order to collect data, online questionnaire was delivered to students at the University of Ha'il between October and December 2023, during the first semester in year 2023–2024. Written informed consent was obtained from all subjects involved in this study. The participants filled in the survey anonymously and voluntarily. A total of 265 students who had experience in using asynchronous e-learning participated in filling the survey. Only six responses were eliminated due to incomplete and invalid responses. Therefore, 259 responses were valid and were processed further for analysis. This study has been reviewed and approved by the Research Ethics Committee (REC) at University of Ha'il dated: 9/10/2023, No. of Research: H-2023-368.

**Table 1 Demographic information.**

| | Demographic characteristics | Frequency | Percent |
|---|---|---|---|
| Gender | Male | 145 | 56.0 |
| | Female | 114 | 44.0 |
| Academic programme | Diploma | 24 | 9.3 |
| | Bachelor's | 215 | 83.0 |
| | Master's | 20 | 7.7 |
| | College of Education | 31 | 12.0 |
| | College of Science | 26 | 10.0 |
| | College of Business Administration | 94 | 36.3 |
| College | College of Computer Science and Engineering | 49 | 18.9 |
| | Applied College | 14 | 5.4 |
| | College of Arts | 45 | 17.4 |
| | Total | 259 | 100.0 |

### *Data analysis*

Two types of statistical analysis were applied: SPSS was used to analyse the respondents' demographic information, and two steps in AMOS were applied to develop a measurement model by CFA and to assess relationships among constructs and test hypothesis by SEM.

## RESULTS

### Demographic information analysis

A total of 259 students who had experience in using asynchronous e-learning participated in the survey. Respondents' demographic information is shown in Table 1. Regarding the respondents' gender, 145 (56.0%) were male students, and 114 (44.0%) were female students. In relation to the respondents' academic programme, the vast majority of the students (215; 83.0%) were enrolled in a bachelor's programme. Just 24 (9.3%) and 20 (7.7%) students were enrolled in diploma and master's programmes, respectively. Most of the participants were studying at the College of Business Administration (94; 36.3%), followed by the College of Computer Science and Engineering (49; 18.9%) and then the College of Arts (45; 17.4%). A small number of students (14; 5.4%) were studying at the Applied College.

### Confirmatory factor analysis

Pooled CFA is the most convenient technique used to develop and validate a measurement model, as it has the power to account for all correlations of constructs and handle measurement errors (*Hair et al., 2010*; *Awang, 2015*). During CFA, validities of construct, convergent and discriminant should be evaluated to develop measurements (*Awang, 2015*). Figure 2 presents CFA output.

Construct validity is met once the model fitness reaches the threshold values suggested in the literature (*Awang, 2015*). The values of the model fitness indices met the recommended levels, as Table 2 presented.

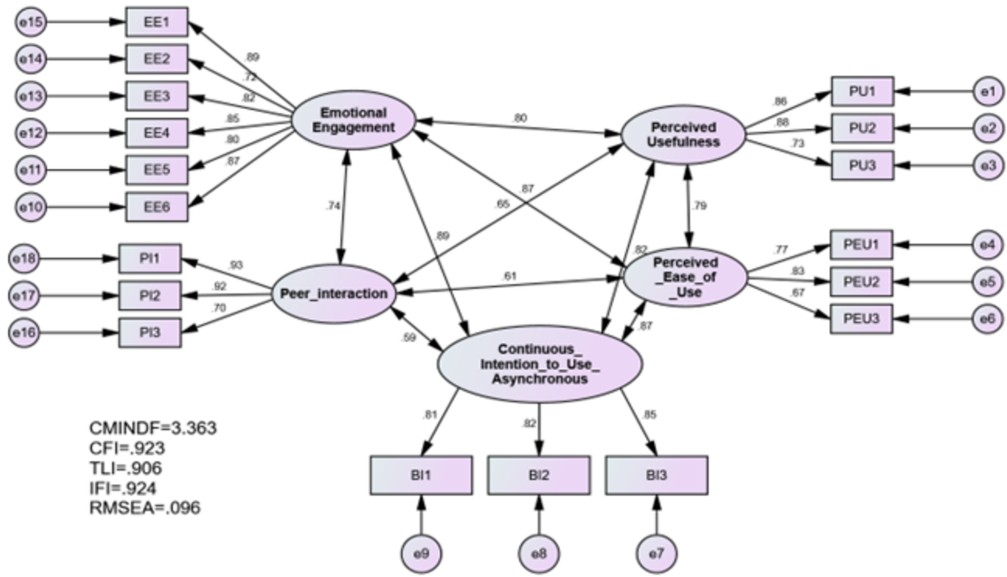

**Figure 2  Confirmatory factor analysis.**

**Table 2  Model fitness.**

| Category | Index | value | Accepted value | Results |
|---|---|---|---|---|
| Absolute fit | "RMSEA" | 0.096 | <0.1 | Accepted (*Mac-Callum, Browne & Sugawara, 1996*) |
| | "CFI" | 0.923 | <0.90 | Accepted (*Awang, 2015*) |
| Incremental fit | "TLI" | 0.906 | <0.90 | Accepted (*Awang, 2015*) |
| | "IFI" | 0.924 | <0.90 | Accepted (*Awang, 2015*) |
| Parsimonious fit | "Chi sq/df" | 3.363 | <3.0 | Accepted (*Hu & Bentler, 1999*) |

**Table 3  CR and AVE.**

| Variable | CR | AVE |
|---|---|---|
| PEU | 0.804 | 0.580 |
| Emotional Engagement | 0.928 | 0.685 |
| Peer Interaction | 0.891 | 0.735 |
| BI | 0.867 | 0.685 |
| PU | 0.865 | 0.683 |

Convergent validity is achieved once composite reliability (CR) score is 0.6 or greater and AVE scores 0.5 or greater (*Awang, 2015*). The CR and AVE values displayed in Table 3 show that the suggested values for convergent validity were met.

**Table 4 Discriminant validity.** The bold values show the square root of AVE. All bold values are greater than other values in their columns or rows.

| Variable | PEU | EE | PI | BI | PU |
|---|---|---|---|---|---|
| PEU | **0.886** | | | | |
| Emotional_Engagement | 0.871 | **0.897** | | | |
| Peer_Interaction | 0.607 | 0.744 | **0.858** | | |
| BI | 0.869 | 0.894 | 0.591 | **0.898** | |
| PU | 0.791 | 0.805 | 0.647 | 0.824 | **0.826** |

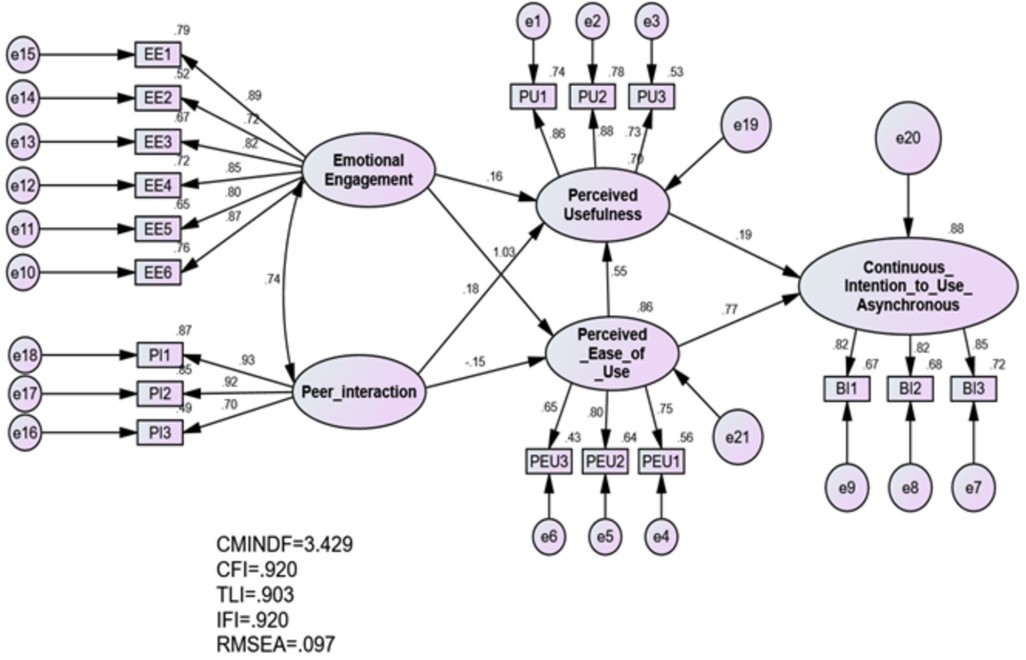

**Figure 3 Standardised estimate.**

Discriminant validity means that each construct is different than other related constructs approved by statistical and empirical standards (*Hair Jr et al., 2021*). For assessing discriminant validity of constructs, the AVE square root should be assessed to compare their correlations in the model (*Fornell & Larcker, 1981*). The bold values in Table 4 show the square root of AVE. All bold values greater than other values in their columns or rows. Therefore, discriminant validity was achieved (*Awang, 2015*).

## Standardised estimate

The SEM presents two kinds of outputs. The standardised estimate is essential for assessing the beta coefficients of variables, factor loading and the *R* square of dependent variables (*Awang, 2015*). The unstandardised estimate is applied for calculating critical ratio for testing hypotheses. The standardised estimate results are presented in Fig. 3.

The *R* square of dependent variable–continuous intention to utilise asynchronous e-learning–was 0.88, which confirms that 88% of continuous intention of using asynchronous

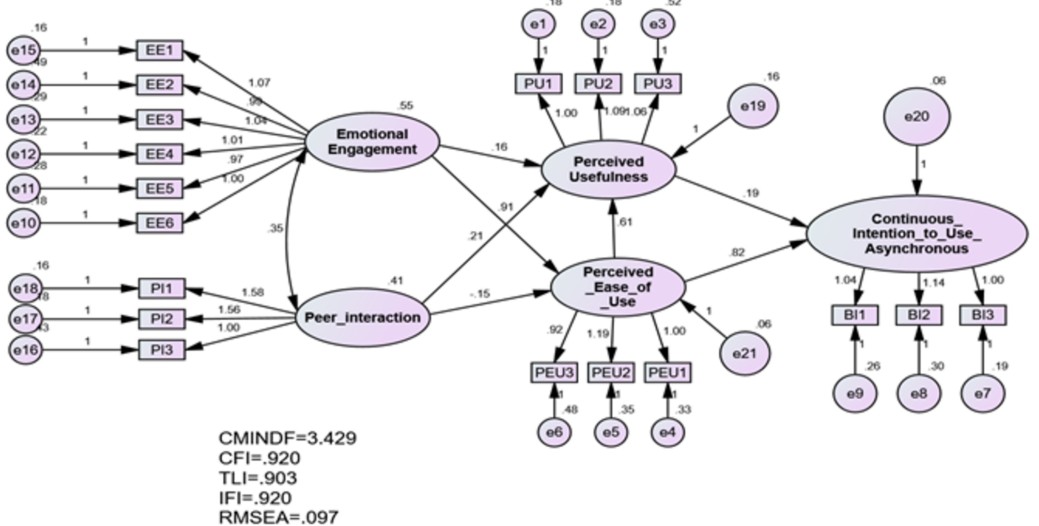

CMINDF=3.429
CFI=.920
TLI=.903
IFI=.920
RMSEA=.097

**Figure 4 Unstandardised estimate.**

e-learning could be explained by other variables. This result confirms the high explanatory power in the proposed model for explaining the phenomenon of students accepting and using asynchronous e-learning and the factors that affect this. *Cohen (1988)* mentioned that, the value of $R$ square less than 0.12 indicates low explanatory power, whereas ranges in the middle of 0.13 and 0.25 refers to medium power and values higher than 0.25 indicate high explanatory power. Thus, this study produced a model with high explanatory power that is able to clearly indicate which factors contribute to continuous intention to utilise asynchronous e-learning.

## Unstandardised estimate

The regression weight which used for assessing critical ratio and testing hypotheses is calculated by applying an unstandardised estimate. Figure 4 presents unstandardised estimate results.

## Results of hypothesis testing

The results show that emotional engagement was insignificant in affecting perceived usefulness ($\beta = 0.156$, $P > 0.05$); therefore, H1 is rejected. However, emotional engagement was significant in affecting perceived ease of use ($\beta = 0.905$, $P < 0.001$). Accordingly, H2 is supported. Furthermore, peer interaction has a significant effect on both PU and PEU ($\beta = 0.206$, $P < 0.05$; $\beta = -0.149$, $P < 0.05$), which supports H3 and H4. Moreover, PEU was significant in affecting PU ($\beta = 0.611$, $P < 0.05$), Therefore, supporting H5. Finally, both PEU and PU significantly affected students' continuous intention to use asynchronous e-learning ($\beta = 0.823$, $P < 0.05$; $\beta = -0.187$, $P < 0.05$). Thus, H6 and H7 are confirmed. Table 5 presents hypothesis testing results.

**Table 5** **Hypothesis results.** Asterisks (***) indicate that the $P$ value $< .001$.

| Variables | | | Estimate | SE | CR | P | Decision |
|---|---|---|---|---|---|---|---|
| Emotional_Engagement | —> | PU | .156 | .246 | 0.635 | .526 | 'Not Significant' |
| Emotional_Engagement | —> | PEU | .905 | .080 | 11.263 | *** | 'Significant' |
| Peer_Interaction | —> | PU | .206 | .096 | 2.159 | .031 | 'Significant' |
| Peer_Interaction | —> | PEU | −.149 | .069 | −2.162 | .031 | 'Significant' |
| PEU | —> | PU | .611 | .253 | 2.420 | .016 | 'Significant' |
| PEU | —> | BI | .823 | .114 | 7.236 | *** | 'Significant' |
| PU | —> | BI | .187 | .092 | 2.026 | .043 | 'Significant' |

**Table 6** **Indirect effect.**

| Variable | BI to asynchronous e-learning | Lower bounds | Upper bounds | P-value |
|---|---|---|---|---|
| Emotional_ Engagement | .878 | 0.680 | 1.071 | .001 |

Furthermore, results confirmed the indirect effect of emotional engagement on students' continuous intention to use asynchronous e-learning ($\beta = 0.878$, $P < 0.05$), the indirect effect value is shown in Table 6.

## DISCUSSION

This study aimed to propose a theorical model and examine the factors which may affect students' continuous intention of using asynchronous e-learning. Specifically, it aimed to explore the effects of emotional engagement and peer interaction along with TAM factors. The findings showed that emotional engagement affected significantly students' continuous intention to use asynchronous e-learning through perceived ease of use. Surprisingly, emotional engagement has an insignificant effect on the perceived usefulness of asynchronous e-learning. Peer interaction has a significant effect on both the perceived ease of use and the perceived usefulness of asynchronous e-learning. Moreover, the TAM factors in the model–PEU and PU–both have direct effects on students' continuous intention to use asynchronous e-learning.

The findings revealed that emotional engagement has a significant effect on students' continuous intention to use asynchronous e-learning through perceived ease of use. These findings are in line with prior studies (*Liang et al., 2021*; *Wang et al., 2015*). The findings showed that emotional engagement affects the perceived ease of use of asynchronous e-learning but not the perceived usefulness of asynchronous e-learning. An explanation for this could be that when students have positive emotions regarding asynchronous e-learning and are willing to use it, asynchronous e-learning becomes easy to use for them but not necessarily useful. Perceiving asynchronous e-learning as easy to use affects positively students' intention to continue using it.

Moreover, findings indicated that peer interaction has a significant effect on students' continuous intention to use asynchronous e-learning and affects both the perceived ease of use and the perceived usefulness of asynchronous e-learning. These findings are in line

with prior studies (*Liu et al., 2010*; *Ros et al., 2015*; *Zhou, Xue & Li, 2022*). This indicates that when students have positive interactions with other students and instructors, they would have a higher chance to perceive asynchronous e-learning as easy and useful, which positively affects their continuous intention to use asynchronous e-learning.

Moreover, the findings revealed that the original TAM constructs–PU and PEU–have significant effects on students' continuous intention to use asynchronous e-learning. PEU of asynchronous e-learning also affects its PU. These findings confirms and in line with prior studies (*Anyim, 2020*; *Hepola, Karjaluoto & Shaikh, 2016*; *Hsu & Lin, 2022*; *Nagy, 2018*). This indicates that students perceive asynchronous e-learning as useful when they first find it easy to use. Therefore, the ease of using asynchronous e-learning needs to be enhanced by providing students with technical tools and training on how to use it. The findings also showed that PEU and PU have direct effects on students' continuous intention to use asynchronous e-learning. This indicates that once learners perceive asynchronous e-learning as easy and useful, they intend to continue using it.

## Implications

This study has several theoretical implications. Few studies have examined the continuous intention to use asynchronous e-learning. *Lu, Pang & Shadiev (2023)* argued for extending TAM and examining the effects of emotional engagement and peer interaction on students' continuous intention to use asynchronous e-learning. It was not yet clear whether these two factors affect students' asynchronous e-learning. This study filled that gap and contributed to the literature by proposing a theoretical model incorporating emotional engagement and peer interaction and examining their effects on students' asynchronous e-learning. To the best of our knowledge, this is the first study that proposes a theoretical model incorporating emotional engagement and peer interaction along with TAM constructs to examine their effects on students' continuous intention to use asynchronous e-learning.

Moreover, the study findings have practical implications. First, it is important to ensure the easiness and usefulness of asynchronous e-learning, as these factors play a big role in students' continuous intention to use asynchronous e-learning. That can be achieved by providing multiple resources for learning, useful content and different technologies. Furthermore, training students on how to navigate and use e-learning materials will make using asynchronous e-learning easy, which will lead to students perceiving it as useful. Second, there is a need to enhance the positive emotional engagement of students in asynchronous e-learning in terms of their interests, values and happiness. Third, instructors should pay attention to students' peer interactions with their instructors and fellow students during asynchronous e-learning and provide collaborative learning activities that require interactions among students, such as discussions on discussion forums or blogs. This will increase students' peer interactions and their continuous intention to use asynchronous e-learning.

## Limitations

Some limitations of this study can be identified. To start with, the study data were self-reported by students and were subjective. Other forms of data collection, such

as observations, need to be taking in consideration for future studies. Furthermore, participants were enrolled in one university in Saudi Arabia. Future studies should include participants from different universities in Saudi Arabia or in other countries for comparing the findings with this study and generalising findings. Finally, this study incorporated two key external factors within TAM: emotional engagement and peer interaction. Future studies should incorporate other important factors, such as the instructional design and accessibility of asynchronous e-learning.

### Funding
This research was funded by University of Ha'il, Fund No 202415. The APC was funded by the authors of the article. The funders had no role in study design, data collection and analysis, decision to publish, or preparation of the manuscript.

### Grant Disclosures
The following grant information was disclosed by the authors:
University of Ha'il:  202415.

### Competing Interests
The authors declare there are no competing interests.

### Author Contributions

- Sultan Hammad Alshammari conceived and designed the experiments, analyzed the data, performed the computation work, prepared figures and/or tables, and approved the final draft.
- Mohammed Habib Alshammari conceived and designed the experiments, performed the experiments, performed the computation work, authored or reviewed drafts of the article, and approved the final draft.

### Ethics
The following information was supplied relating to ethical approvals (i.e., approving body and any reference numbers):

This study has been reviewed and approved by the Research Ethics Committee (REC) at University of Ha'il dated: 9/10/2023. Approval Number: H-2023-368.

### Data Availability
The raw data is available in the Supplemental File.

### Supplemental Information
Supplemental information for this article can be found online at http://dx.doi.org/10.7717/peerj-cs.1990#supplemental-information.

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
