# Peer review of "Modelling the effects of emotional engagement and peer interaction on the continuous intention to use asynchronous e-learning"

_PeerJ Computer Science, doi:10.7717/peerj-cs.1990_

## Round 0.1 · original submission · Major Revisions

According to the reviewers, the manuscript must be revised.

**Language Note:** PeerJ staff have identified that the English language needs to be improved. When you prepare your next revision, please either (i) have a colleague who is proficient in English and familiar with the subject matter review your manuscript, or (ii) contact a professional editing service to review your manuscript. PeerJ can provide language editing services - you can contact us at [email protected] for pricing (be sure to provide your manuscript number and title). – PeerJ Staff

Reviewer 1 ·

Basic reporting

The paper, "Modelling the effects of emotional engagement and peer interaction on the continuous intention to use asynchronous e-learning," aims to explore how emotional engagement, peer interaction, and the Technology Acceptance Model (TAM) constructs influence students' continuous intention to use asynchronous e-learning.

Experimental design

Utilized a survey with 259 student participants, the study employs Structural Equation Modeling (SEM) for analysis. It finds significant effects of emotional engagement on perceived ease of use (but not on perceived usefulness) and of peer interaction on both perceived usefulness and ease of use. TAM constructs also significantly affect continuous intention to use asynchronous e-learning.

Validity of the findings

N/A

Additional comments

Comments.
1. The study's sample is limited to one university in Saudi Arabia, which may affect the generalizability of the findings to other contexts or cultures. To enhance generalizability, future research should include a more diverse sample from multiple institutions and possibly across different countries.
2. Relying solely on self-reported questionnaires could introduce bias, as responses may not accurately reflect actual behaviours or intentions. Incorporating qualitative data through interviews or observations could provide deeper insights into the reasons behind the quantitative findings and uncover additional factors influencing e-learning engagement.
3. The study focuses on emotional engagement and peer interaction but may overlook other factors that could influence asynchronous e-learning usage, such as instructional design, content quality, or accessibility issues. Investigate other potential factors, such as instructional design, content relevance, or technology accessibility, to understand their impact on students' continuous intention to use asynchronous e-learning more comprehensively.
4. Conducting a longitudinal study could help in understanding how students' intentions and behaviours change over time, providing a more dynamic view of the factors influencing asynchronous e-learning usage.

Reviewer 2 ·

Basic reporting

This study proposed a theoretical model aiming to examine the effects of external factors, emotional engagement, peer interaction, and two constructs of the Technology Acceptance Model (TAM), on students' continuous intention to use asynchronous e-learning. A total of 259 students participated in a survey. The data were analyzed using two steps in SEM AMOS. Confirmatory factor analysis was conducted to assess model measurement, and structural equation modeling was applied to assess relationships among constructs and test hypotheses. The results showed that emotional engagement had a significant effect on students' continuous intention to use asynchronous e-learning through the perceived ease of use of asynchronous e-learning and not through its perceived usefulness.

The authors should provide an explanation of asynchronous e-learning before delving into the details of the study. Additionally, while the paper appears to be novel and interesting, there are several questions regarding the contribution of computer science to this study.
The manuscript lacks details on models, algorithms, systems, or any application of computer science. It's unclear how emotional engagement was measured,
how hypotheses were formulated,
how the system works,
and whether any machine learning models were used.

Experimental design

As per basic reporting.

Validity of the findings

As per basic reporting.

---

## Round 0.2 · accepted · Accept

Based on the reviewers, the manuscript can be accepted.

Reviewer 1 ·

Basic reporting

N/A

Experimental design

N/A

Validity of the findings

N/A

Additional comments

The authors justified the comments and addressed some of them.

Reviewer 2 ·

Basic reporting

The authors addressed all previous concerns and hence accepted.

Experimental design

Nil

Validity of the findings

The authors addressed all previous concerns and hence accepted.